# Mud Spectral Characteristics from the Lusi Eruption, East Java, Indonesia Using Satellite Hyperspectral Data

Stefania Amici [1,*], Maria Fabrizia Buongiorno [1], Alessandra Sciarra [1] and Adriano Mazzini [2,3]

1   Istituto Nazionale di Geofisica e Vulcanologia, via di Vigna Murata 605, 00143 Rome, Italy;
    fabrizia.buongiorno@ingv.it (M.F.B.); alessandra.sciarra@ingv.it (A.S.)
2   Department of Geosciences, University of Oslo, Box 1047 Blindern, 0316 Oslo, Norway;
    adriano.mazzini@geo.uio.no
3   Institute for Energy Technology, 2007 Kjeller, Norway
*   Correspondence: stefania.amici@ingv.it

**Abstract:** Imaging spectroscopy allows us to identify surface materials by analyzing the spectra resulting from the light–material interaction. In this preliminary study, we analyze a pair of hyperspectral cubes acquired by PRISMA (on 20 April 2021) and EO1- Hyperion (on 4 July 2015) over the Indonesian Lusi mud eruption. We show the potential suitability of using the two sensors for characterizing the mineralogical features in demanding "wet and muddy" environments such as Lusi. We use spectral library reflectance spectra like Illite Chlorite from the USGS spectral library, which are known to be associated with Lusi volcanic products, to identify minerals. In addition, we have measured the reflectance spectra and composition of Lusi sampled mud collected in November 2014. Finally, we compare them with reflectance spectra from EO1-Hyperion and PRISMA. The use of hyperspectral sensors at improved SNR, such as PRISMA, has shown the potential to determine the mineral composition of Lusi PRISMA data, which allowed the distinction of areas with different turbidities as well. Artifacts in the VNIR spectral region of the L2 PRISMA reflectance product were found, suggesting that future work needs to take into account an independent atmospheric correction rather than using the L2D PRISMA product.

**Keywords:** EO1-Hyperion; PRISMA mission; imaging spectroscopy; Lusi; Illite

## 1. Introduction

Imaging spectroscopy allows us to characterize materials by measuring unique spectra (reflectance, absorption, and transmission) that result from the interaction of light with the surface. More specifically, hyperspectral sensors operating in the VNIR (Visible Near Infrared) and SWIR (Short Wave Infrared) allow us to derive reflectance spectra. These spectra can provide indications about composition by identifying unique features known as "fingerprint" or spectral indicators [1]. Imaging spectroscopy from space is used in many applications, including (but not limited to) vegetation [2], agriculture, [2] food security [2], hydrology cryosphere [2,3], environment degradation, [2] natural hazards [2], raw materials and geological studies [2–5].

Regarding the mineral and geological characterization of Earth's surface, over the past 35+ years, hyperspectral data have mainly been used to identify surface materials covering diverse areas. Several airborne sensors have been used, including the pioneering Australian HyMap [6], Airborne Visible and Infrared Imaging Spectrometer (AVIRIS) [7,8], Airborne Imaging Spectrometer (AIS) [8], Airborne Hyperspectral Scanner (AHS) [8], Airborne Imaging Spectrometer for different Applications (AISA) [8], Airborne Prism Experiment (APEX) [8], Airborne Reflective Emissive Spectrometer (ARES), Digital Airborne Imaging Spectrometer (DAIS-7915) [8], Hyperspectral Mapper (HyMap) [8], Hyperspectral Digital Imagery Collection Experiment (HYDICE) [8], Multispectral Infrared and Visible Imaging Spectrometer (MIVIS) [8], and the Operational Modular Imaging Spectrometer (OMIS) [8].

To note that AVIRIS, a NASA-research system, is mainly used within USA due to severe restrictions to its use outside. Since 1995, the above-mentioned systems have been used to collect hyperspectral VNIR-SWIR data for mineral and geological characterization for "small" and/or "large areas" surveys (including surveys on the scale of countries and continents).

The use of hyperspectral sensors from space [9,10] has been limited by the availability of the sensors (i.e., EO1-Hyperion operating between 2000–2017). The advent of new imaging spectrometers onboard satellites such as PRISMA (launched in 2019), EnMap (launched in 2022), and on the International Space Station as HSUI (2019) and EMIT (2022), are offering opportunities to study surface composition and gas emissions from point sources. For example, PRISMA has successfully characterized minerals in semi-arid environments [11,12].

Here, we describe the potential use of space imaging spectroscopy to characterize the composition of mud expelled from the Lusi eruption site in Indonesia. Lusi (contraction of LUmpur SIdoarjo) is a sediment-hosted geothermal system that appeared in the Sidoarjo village (Northeast Java, Indonesia) [13–18]. The eruption occurs along the Watukosek fault system, which also hosts other mud volcanoes further to the NE of the Island [19–24]. The Lusi inception occurred on 29 May 2006 and continues today with the eruption of copious (up to 180,000 m$^3$/day) amounts of mud, clasts, water, oil, and gas [18]. The analysis of the mud and clasts reveals the presence of Smectite, Kaolinite, Illite, and minor Chlorite [14,22,25,26]. Geochemical analyses reveal that the main sources of the erupted Lusi water are a mixture of hydrothermal and meteoric fluids trapped with seawater and fluids released from clay illitization [16–18,24,26]. In this study, we combine spectral library and reflectance spectra measured in the laboratory of Lusi mud samples as references to analyze the hyperspectral cubes acquired by EO-1 Hyperion (2015) and PRISMA (2021) over Lusi and show the evolution of the Lusi mud emissions. We describe the method used to select reference spectra and compare laboratory vs. EO1_Hyeperion and PRISMA spectra to identify minerals present in their images. We discuss the results in terms of the suitability of PRISMA to identify the mineral composition of the Lusi mud.

## 2. Materials and Methods

### 2.1. Study Area: Geological Setting

Lusi is situated in the NE Java backarc basin and is located ~13 km to the NE of the Penanggungan volcano [15,27]. The local stratigraphy has been constrained from borehole data complemented by regional studies, seismic data interpretations, and analysis of erupted clasts [14,22,26–28]. The main identified units (top to bottom) include altered sand, shale, and clay recent sediments and the Pucangan Formation (Pleistocene), Bluish Grey clay from the Upper Kalibeng Formation (Pleistocene), volcanoclastics from the Upper Kalibeng Formation (Pleistocene), marls from the Tuban Formation (Miocene), carbonates from the Kujung-Prupuh Formation (Oligocene-Miocene) and shales from the Ngimgbang Formation (Eocene-Oligocene). The sediments that erupted from Lusi contain a mixture of all these Formations.

### 2.2. Hyperspectral Satellite Data

#### 2.2.1. EO1-Hyperion

EO1-Hyperion: the instrument was launched on 21 November 2000 on board the Earth Observing-1 (EO-1) satellite as NASA's New Millennium Program Earth Observing. It was meant to be a one-year technology demonstration/validation mission [29,30]. However, the mission was extended several times and finally decommissioned in March 2017.

The instrument acquires 220 contiguous spectral bands, covering the range from 400 nm to 2500 nm at a ground resolution of 30 m. It is a push broom instrument with a spatial resolution of 30 m for all bands with a scene width of 7.7 km, a standard scene length of 42 km, and an optional increased scene length of 185 km (Table 1). It has a single telescope and two spectrometers, one visible/near-infrared (VNIR) (with CCD detector

array) and one short-wave infrared (SWIR) (HgCdTe detector array) [31] EROS Archive—Earth Observing One (EO-1)-Hyperion. The image used for this study was acquired on 4 July 2015 (Figure 1a). The L1 was atmospherically corrected using the QUAC [32] available in ENVI 5.5 [33], which provides good performances in cloud-free and no cloud-shadowed scenes, with images that contain diverse materials such as soil, vegetation, and manmade structures, as in our case [34]. The QUAC returns apparent reflectance integer data, with pixel values ranging from 0 to 10,000 (representing 0 to 100% reflectance) [33,35]. Empirical Flat Field Optimal Reflectance Transformation (EFFORT) correction tool available under ENVI was used to remove most of the systematic noise.

**Table 1.** EO1-Hyperion and PRSIMA sensors characteristics.

| Specification | EO1-Hyperion | PRISMA |
|---|---|---|
| Swath with | 7.75 km | 30 km |
| Spectral channels | VNIR (70 channels, 356–1058 nm), SWIR (172 channels, 852–2577 nm) | VNIR (66 channels, (400–1010 nm) SWIR (174 channels, 920–2505 nm) PAN 1 channel |
| Spectral bandwidth | 10 nm | VNIR (9–13 nm) SWIR (9–14.5 nm) |
| Signal-to-Noise Ratio (SNR) | 161 (550 nm); 147 (700 nm); 110 (1125 nm); 40 (2125 nm) | >160 (>450 at 650 nm) >100 (>360 at 1550 nm; >240 (PAN) |
| Altitude | 705 km | 615 km |
| Revisit time | 16 days | 29 days (nadir) and 7 days (off nadir) |
| Absolute radiometric accuracy | N/A | Better than 5%, |

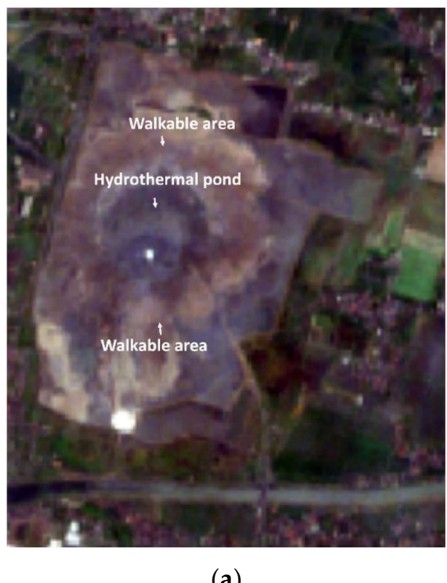
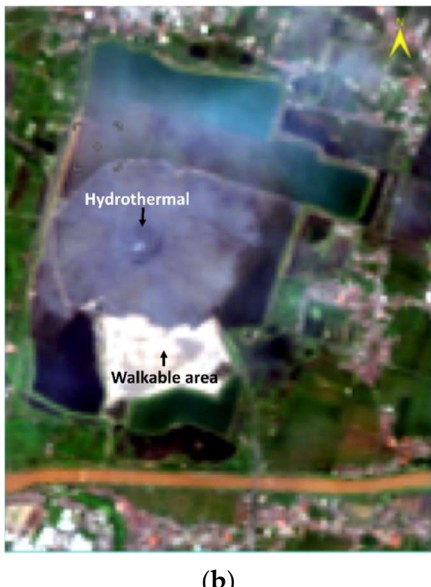

(**a**)                 (**b**)

**Figure 1.** RGB color composite in the visible spectral bands around the Lusi area acquired by satellite. (**a**) EO1-Hyperion on 4 July 2015; the brownish area is walkable and accessible with dry mud breccia. The active crater (dark grey) creates a large hydrothermal pond 65and some water funnels; (**b**) PRISMA on data 20 April 2021, the white areas are walkable while large grey areas around the crater appear very unsettled (not walkable).

2.2.2. PRISMA

PRecursore IperSpettrale della Missione operativA (PRISMA) is a hyperspectral mission by the Italian Space Agency (ASI). PRISMA is a push broom instrument with a 30 km wide imaging swath composed of two cameras: (1) the imaging spectrometer (hyperspectral camera) that operates in the spectral range spanning between 400–2500 nm with a spectral resolution ≤ 12 nm and at Ground Sampling Distance (GSD) of 30 m/pixel and

(2) a Panchromatic camera that acquires the same area at 5 m/pixel [36,37]. In Table 1 are reported PRISMA characteristics.

The Italian Space Agency releases PRISMA data at different levels of processing. In this work, we used the L2D level, atmospherically corrected at-surface reflectance product [36]. The image acquired on 20 April 2021 (Figure 1b) has very limited cloud cover and a maximum temperature of 30 °C in a day. The VIS (66 channels) and SWIR (174 channels) cubes have been combined by using the stack layer option in the ENVI 5.6 software, which also allows the removal of the overlapping wavelength.

*2.3. Reference Reflectance Spectra*

2.3.1. Spectral Library Data

Spectral libraries result from collected reflectance spectra of minerals measured in laboratory under controlled environment and following specific protocols with associated information (i.e., composition, sample picture, etc.). These libraries represent the reference (truth) used to compare measurements realized in situ for geological validation of satellite data. We have been exploring the literature to find mineral types that match those identified at Lusi or in other similar settings that have been found on Lusi.

Hydrothermal water chemistry and bedrock lithology can be inferred from the types of minerals on the land surface [38]. The observed altered rock identified from imaging spectroscopy data can be linked with the underlying geological process and its associated mineral trends with fracture systems [38]. The mud erupted at Lusi, which emerges from deep, contains various types of clay minerals, including Smectite, Kaolinite, Illite, and minor Chlorite [25]. Illite is a secondary mineral precipitate that belongs to a group of mica-type clay minerals characterized by a micalike sheet structure and is poorly crystallized. The space between its sequence of layers is occupied by poorly hydrated potassium cations, which are responsible for the absence of and/or poor swelling ability, especially compared to Smectites.

The presence of Illite is typical of sedimentary basins where Smectite to Illite conversion occurs. Analyses on side well cores from the BJP1 borehole (close to Lusi) revealed that Illite-Smectite conversion (a dehydration reaction) occurs in the sediments of the Bluish Grey clays from the Upper Kalibeng Formation [14].

Clay minerals are characterized by diagnostic absorption features near 1400 nm (caused by OH overtones), and 1900 nm (overtones caused by water molecules) due to Al-OH combination tones at 2120, 2209, 2133, 2225 nm, 2250 nm) [39]; some weaker absorptions are present in the 2300–2500 nm range due to presence of Fe- or Mg-OH and Iron at 477 nm, 556, 693, [40,41]. We have explored USGS spectral library version 7 (available at https://www.usgs.gov/data/usgs-spectral-library-version-7-data (accesed on 24 april 2024) for full compositional description) and accessed within ENVI 6.6.1 as USGSV6mineralbeckman 430 and USGS V6 mixtureV6 and USGSV7liquidsASDfr. The spectra used in the study were measured by using the Analytical Spectral Device spectrometer at full range standard resolution in the spectral range 350 to 2500 nm (old code W1R1F, recently re-coded as ASDF) and the Beckman spectrophotometer 200 to 3000 nm (old code W1R1B recently re-coded as BECK) [42,43].

The specific reflectance spectra of five Illite, three Calcite, and two Chlorite spectra have been found in the USGS V6mineralbeckman 430 spectral library, and their reflectance spectra are plotted in Figure 2.

To compare the spectra from both satellites vs. the in situ spectra, we first applied the Empirical Flat Field Optimal Reflectance Transformation (EFFORT) tool of ENVI 5.7 to reduce systematic noise. For EO1-Hyperion we selected five segments to avoid the following wavelength intervals, 922–962 nm, 1104–1154 nm, 1326–1487 nm, 1790–1991 nm, characterized by $O_2$, $CO_2$, and water vapor absorptions. For PRISMA data, we also selected five segments avoiding the four wavelength intervals: 912–978 nm, 1131–1152 nm, 1328–1492 nm, and 1784–1967 nm. The polynomial order was set to 10 for all the segments.

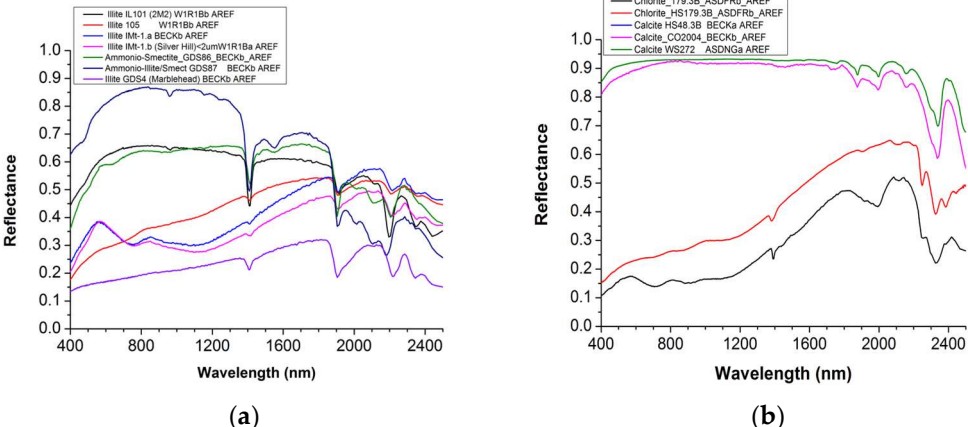

**Figure 2.** (**a**) Illite, Ammonium –Smectite and Ammonium Smectite reflectance spectra. (**b**) Chlorite, Calcite spectra from spectral libraries. Sequence of letters and numbers after the mineral name allows us to identify the sample. Data source USGS spectral library.

Secondly, we applied the continuum removal to identify minerals, and finally, a Spectral Angle Mapping (SAM) supervised classification was implemented in ENVI Software 5.6software to create a map.

### 2.3.2. Lusi Mud Spectra

Two samples were collected from the Lusi site (Lat 7.52953 Lon 112.71012) during fieldwork on 30 November 2014. One mud sample *mud from the stream* was collected from one of the crater outflow streams and represented freshly discharged sediments. The second sample is surface *dry mud* characterized by some superficial alterations, such as white drying marks. Figure 3 shows the location where the samples were taken.

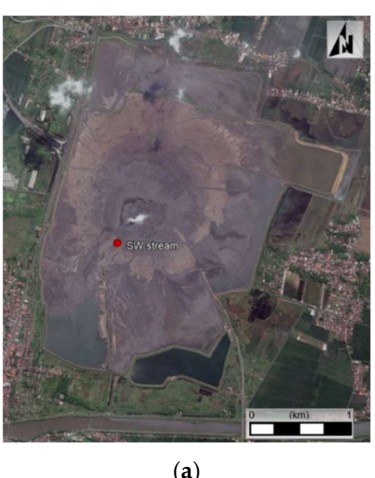

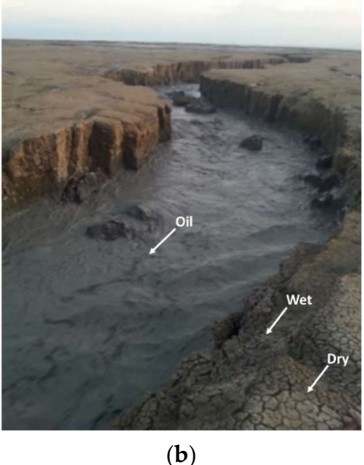

**Figure 3.** (**a**) Satellite image (from Google Earth) of Lusi and SW stream location (red dot). (**b**) Detail of SW stream showing trace of oil in the freshly erupted fluids and the wet and dry mud surfaces.

The samples were split into two parts: one used to measure the reflectance spectra and the other for chemical analysis (see Section 2.4).

The laboratory reflectance measurements were conducted using an ASD Fieldspec by Analytical Spectral Device, (ASD) operating between 350–2500 nm using the contact probe. The *mud from the stream* was dried before measuring the reflectance (Figure 4). The measurements were repeated three times at different points of the sample. Reflectance was measured in five different points in the *dry mud* sample (Figure 3) due to the larger size of the sample (about 5 cm diameter).

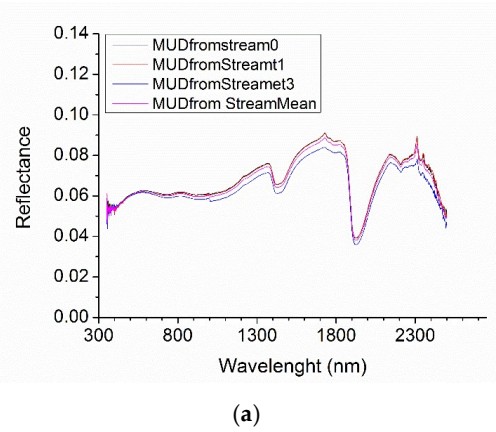
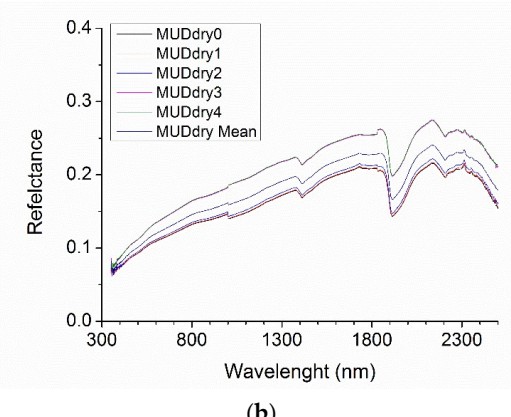

(**a**)  (**b**)

**Figure 4.** (**a**) Reflectance spectra acquired in diverse points of the stream mud sample *mud from the stream* and their average; (**b**) Reflectance spectra from the *mud dry* sample and their average. Note: a different scale has been used on the Y axes to highlight absorption features.

### 2.4. Chemical Analysis

The remaining part of the two mud samples are untreated sediments manually sieved through a 2 mm mesh. To remove interstitial water, the samplers were dried in a stove at 60 °C for 24 h and successively powdered using an Agata mortar. The mineralogical analyses were carried out by means of XRPD Analyses with a Philips PW1860/00 diffractometer. An open-source software (Qualx 2.0) was used for the qualitative and semi-quantitative identification of the minerals. Main cations and trace elements were measured using an X-Series Thermo-Scientific spectrometer (ICP-MS) [44]. Analyses were executed with a Thermo Electron Corporation Xeries spectrometer and a collision/reaction cell (CCTED) for the reduction/exclusion of main polyatomic and isobaric interferences.

### 2.5. Examination of Hyperspectral Sensors Reflectance Spectra

The spectral analysis was conducted for both EO1-Hyperion and PRISMA using the following workflow:

- Visual inspection of the volcanic area as mapped by the two hyperspectral sensors was used to explore spectra in both walkable and not walkable areas;
- Spectra comparison between satellite and library/in situ.

We found three different kinds of spectral behavior in the EO1-Hyperion scene. Figure 5 shows an example of a single spectrum acquired in four different areas of the eruption site.

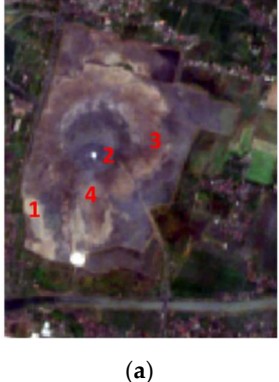
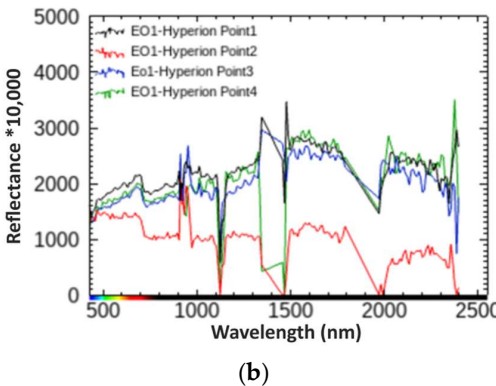

(**a**)  (**b**)

**Figure 5.** (**a**) RGB color composite of Lusi area mapped by EO1 Hyperion on 20 July 2015. Numbers from 1 to 4 indicate the location of areas showing distinctive spectral behavior (**b**) Reflectance spectra of points 1–4.

By exploring the PRISMA imagery acquired six years later, we have found five distinct types of reflectance spectra corresponding to different locations in the volcanic area (Figure 6).

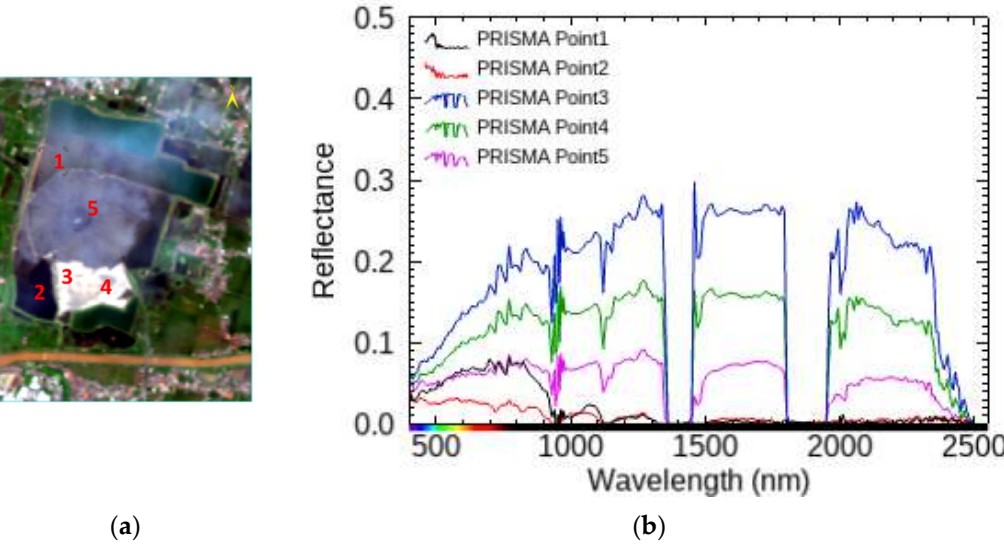

(**a**)                                                          (**b**)

**Figure 6.** (**a**) RGB color composite of L area mapped by PRISMA on 20 April 2021. Numbers from 1 to 5 indicate the location of areas showing distinctive spectral behavior. (**b**) Reflectance spectra showing different characteristics of points 1–5 in (**a**).

## 3. Results

In this session, we discuss the results of the multiple-scale approach spectra, comparing the spectral library against the spectra of the two samples and the satellite-derived ones.

### 3.1. Comparison Lusi Samples vs. Spectral Library

The characteristic spectral profiles allow mineral identification by simple feature recognition against reference spectra [4,44,45]. The best fit to a library spectrum usually corresponds to the spectrally dominant material. The reflectance spectra from the reference library have been compared with the reflectance spectra of the *mud from the stream* sample and the *dry mud* measured in the laboratory.

The spectra of the *mud from the stream* (Figure 7a) are in good agreement with that from IMt-1a reflectance (Figure 5a). Although the reference spectrum and the measured ones have diverse scales, the identification of minerals uses spectral shape and features rather than reflectivity values to retrieve the minerals. In addition to the overall agreement in the shape of the curves, a correspondence between distinctive features, such as the electronic $Fe^{2+}$ (620–650 nm), $Fe^{3+}$ (830–970 nm), and Illite the vibrational 2.215 nm and 2.345 nm can be seen (Figure 7a). The feature observed between 1725–1760 nm in the *mud from the stream* sample is typical of hydrocarbons. ICP-MS analyses can detect most elements from Li to U, except C, N, O, Cl, Br, I, and S. For these reasons, the analyses cannot be compared with those from the library regarding the presence of hydrocarbons. Indeed, the presence of hydrocarbons is directly visible in the streams since copious amounts of oil are discharged from the crater, as highlighted by aerial images and organic geochemistry analyses [15,22,46] Illite IMt-1.a sample belongs to the phyllosilicate mineral type and the hydrated mica clay minerals group. It was collected in Silver Hill, Montana, and its spectrum was initially published by [47], followed by the EM analysis by Gregg A. Swayze, Branch of Geophysics, USGS, Denver.

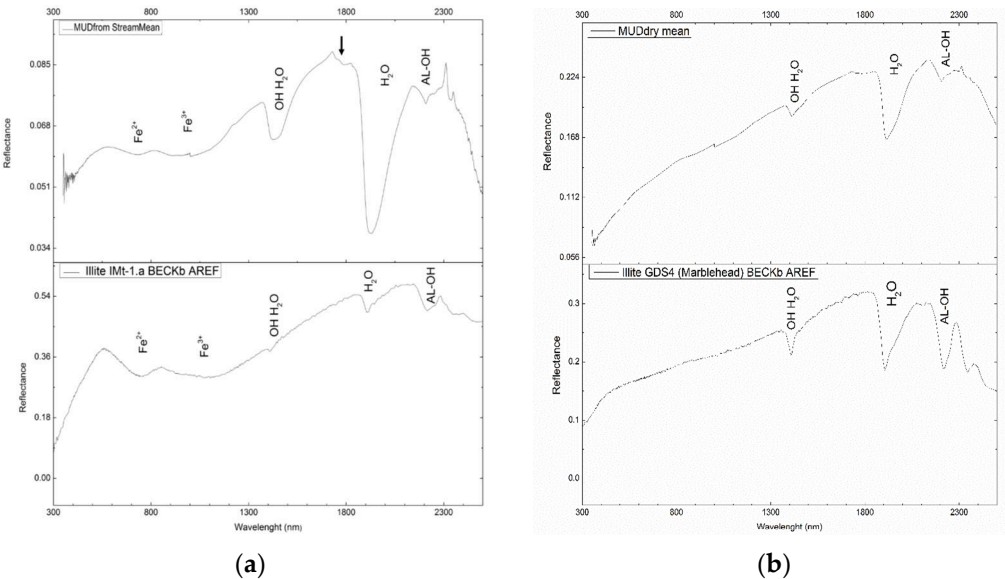

|                | (a)            |                | (b)            |

**Figure 7.** (**a**) Illite I Mt-1 is plotted against the *mud from the stream* laboratory measurements. The typical absorption bands of $Fe^{2+}$, $Fe^{3+}$, OH, $H_2O$, and AL-OH are shown, while the blue arrow highlights the position of hydrocarbon absorption features. (**b**) Illite GDS4 spectral shape and absorbing bands are compared against the *dry mud* measurements. The absorption bands OH, H2O, and AL-OH are shown in the graphs. The absorption bands' positions match in both curves.

The *dry mud* spectra matched very well with the reference Illite Marblehead Figure 7b. In Tables 2 and 3 are reported the chemical measurements.

**Table 2.** Chemical composition of the "mud wet" sample and the Illite IMt-1.a W1R1Bb AREF.

| Oxide ASCII | Amount MUD from Stream | Amount Illite IMt-1.a | Weight Percent, % | Oxide Html |
|:---:|:---:|:---:|:---:|:---:|
| $SiO_2$ | 52.60 | 52.10 | wt% | $SiO_2$ |
| $TiO_2$ | 0.81 | 0.79 | wt% | $TiO_2$ |
| $Al_2O_3$ | 18.28 | 21.90 | wt% | $Al_2O_3$ |
| $Fe_2O_3$ | 7.71 | 6.44 | wt% | $Fe_2O_3$ |
| MnO | 0.14 | Less than 0.02 | wt% | MnO |
| MgO | 2.82 | 2.39 | wt% | MgO |
| CaO | 3.73 | 1.07 | wt% | CaO |
| $Na_2O$ | 3.65 | 0.30 | wt% | $Na_2O$ |
| $K_2O$ | 1.50 | 7.84 | wt% | $K_2O$ |
| $P_2O_5$ | 0.11 | 0.10 | wt% | $P_2O_5$ |
| LOI | 8.64 | 6.91 | wt% | LOI |
| Total | 99.99 | 99.56 | wt% | Total |

The geochemistry of the clay fraction of the sampled mud is dominated by $SiO_2$ and $Al_2O_3$, with an average weight of 53% and 18%, respectively. Some other major oxides are also relatively high, including $Fe_2O_3$, $Na_2O$, and $K_2O$, with a ratio of $Na_2O/K_2O$ of greater than 1.9. As shown in Table 2, the oxide contents of the Lusi mud indicate Illite and Smectite-rich Clay.

In particular, the Smectite group of clays has a structure that is similar to that of Illite but can also have significant amounts of Mg and Fe, which replace each other in the octahedral layers. In fact, Smectites can be both dioctahedral and trioctahedral, and one species differs from another precisely due to variations in the chemical composition, which involve the replacement of Al with Si in the tetrahedral cationic sites and Al, Fe, Mg, and Li in the octahedral cationic sites. Moreover, the Smectite group has the ability for $H_2O$ molecules to be absorbed between its strata, causing the volume of the minerals to increase

when they come in contact with water. In terms of composition, the *mud from the stream* sample has $SiO_2$ (52.60), $TiO_2$ (0.81), $Al_2O_3$ (18.20), and $Fe_2O_3$ (7.71) values comparable to the Illite IMt-1a BECKb AREF composition ($SiO_2$ = 52.10, $TiO_2$ = 0.79, $Al_2O_3$ = 21.90, $Fe_2O_3$ = 6.44). In terms of composition, the *mud dry* sample $SiO_2$ (53.71), $TiO_2$ (0.81), and $Al_2O_3$ (18.60) values are comparable to the Illite GDS4 Marblehead composition ($SiO_2$ = 51.62, $TiO_2$ = 0.92, $Al_2O_3$ = 23.96). The spectra from the sample are consistent with minerals associated with Lusi extruded materials.

**Table 3.** Chemical composition of the "*mud dry*" sample and the Illite GDS4 Marblehead.

| Oxide ASCII | Amount MUD Dry | Amount Illite GDS4 Marblehead | Weight Percent, % | Oxide Html |
|---|---|---|---|---|
| $SiO_2$ | 53.71 | 51.62 | wt% | $SiO_2$ |
| $TiO_2$ | 0.81 | 0.92 | wt% | $TiO_2$ |
| $Al_2O_3$ | 18.60 | 23.96 | wt% | $Al_2O_3$ |
| $Fe_2O_3$ | 7.95 | 1.63 | wt% | $Fe_2O_3$ |
| FeO | - | 0.29 | | FeO |
| MnO | 0.13 | 0.01 | wt% | MnO |
| MgO | 2.91 | 3.83 | wt% | MgO |
| CaO | 3.30 | 0.74 | wt% | CaO |
| $Na_2O$ | 3.07 | 0.47 | wt% | $Na_2O$ |
| $K_2O$ | 1.57 | 8.12 | wt% | $K_2O$ |
| $P_2O_5$ | 0.12 | 0.09 | wt% | $P_2O_5$ |
| $H_2O^+$ | - | 5.00 | wt% | $H_2O^+$ |
| $H_2O^-$ | | 2.9 | wt% | $H_2O^-$ |
| LOI | 7.83 | 6.91 | wt% | LOI |
| Total | 100.00 | 99.41 | wt% | Total |

*3.2. Comparison Satellite vs. Spectral Library*

In Figure 8b, the EO1-Hyperion spectrum in point 1 (magenta), which appears to be a walkable area from the satellite, is compared against the Illite spectrum. The Calcite spectra completely mismatched the EO1-Hyperion spectra and have not been plotted. The in situ spectra have been plotted as well. The EO1-Hyperion data are very noisy (many positive and negative spikes) due to the low Signal to Noise Ratio (SNR) with a better performance in the spectral window between 450–900 nm. In this spectral window, the $Fe^{2+}$ can be noticed. A position that is aligned with the Illite Mt1a and Mt1-b and with Illite GDS between 1600 and 2400 nm is shown.

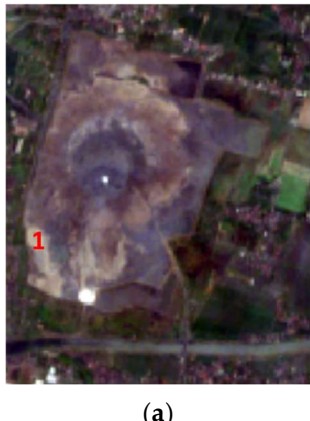

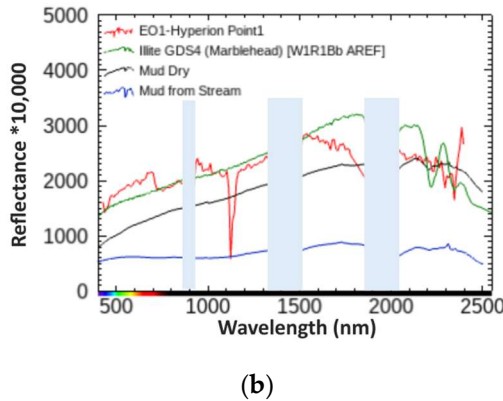

(**a**)                           (**b**)

**Figure 8.** (**a**) True color EO1-Hyperion data color composition. The EO1-Hyperion single pixel spectrum was extracted by the pixel corresponding to point 1, Lat 7°31′47.79″ S, Lon 112°421′28.85″ E, and (**b**) compared vs. the spectral library of two Chlorite spectra and three Illite spectra. Noisy

EO1-Hyperion bands due to the water vapor effect are masked in light blue. $Fe^{2+}$ absorption band can be identified in the EO1-Hyperion spectrum. The geochemistry of the clay fraction of the sampled mud is dominated by $SiO_2$ and $Al_2O_3$, with an average weight of 53% and 18%, respectively. Some other major oxides are also relatively high, including $Fe_2O_3$, $Na_2O$, and $K_2O$, with a ratio of $Na_2O/K_2O$ of greater than 1.9. As shown in Table 2, the oxide contents of the Lusi mud indicate Illite and Smectite-rich Clay.

Figure 9 shows the EO1-Hyperion spectrum acquired close to the Lusi crater (point 2 in red) compared against the Illite spectrum and the spectra from the samples.

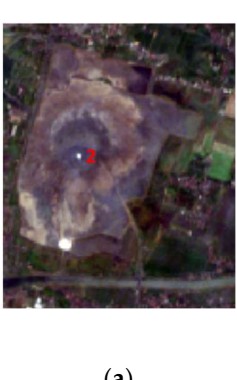

(**a**)

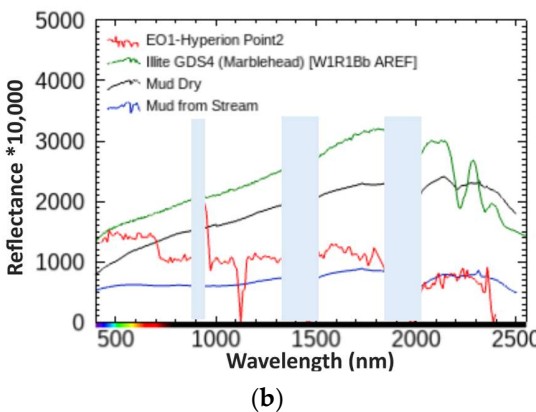

(**b**)

**Figure 9.** (**a**) EO1-Hyperion scene true color composition. (**b**) The EO1-Hyperion single pixel spectrum was extracted by the pixel corresponding to point 2 Lat 7°31′26.21″ S, 112°42′54.18″ E and compared vs. Illite spectral library and the samples. Noisy EO1-Hyperion bands due to water vapor effect are masked in light blue.

Although the EO1-Hyperion shows few positive and negative spikes due to the noise associated with the data, no good agreement in terms of reflectance shape is visible throughout the spectrum, which appears to be similar to that observed in water-dominated spectra such as in the Lusi eruption site.

Figure 10 shows the comparison of the spectral library spectra vs. the EO1-Hyperion spectrum in point 3 (red). The spectral shape is close to the Marblehead spectrum. However, despite the spikes, a weak $Fe^{2+}$ absorption band is distinguishable (about 700 nm). The lower reflectance values are nevertheless comparable, and a resulting flattened spectrum is expected due to a mixing effect of Illite and fluids.

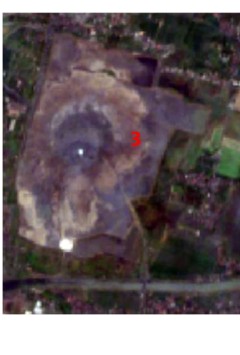

(**a**)

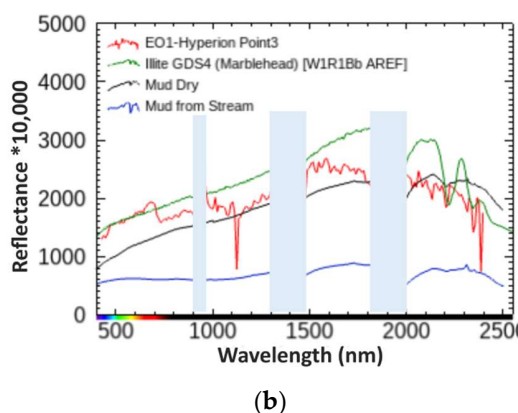

(**b**)

**Figure 10.** (**a**) EO1-Hyperion data natural color composition. (**b**) The EO1-Hyperion single pixel spectrum was extracted by the pixel corresponding to point 3 Lat 7°31′13.44″ S, 112°43′14.66″ E and compared vs. Illite spectrum and the samples. Noisy EO1-Hyperion bands, due to the water vapor effect are masked in light blue.

Figure 11 shows the spectrum acquired in point 4. The Fe⁺ absorption.

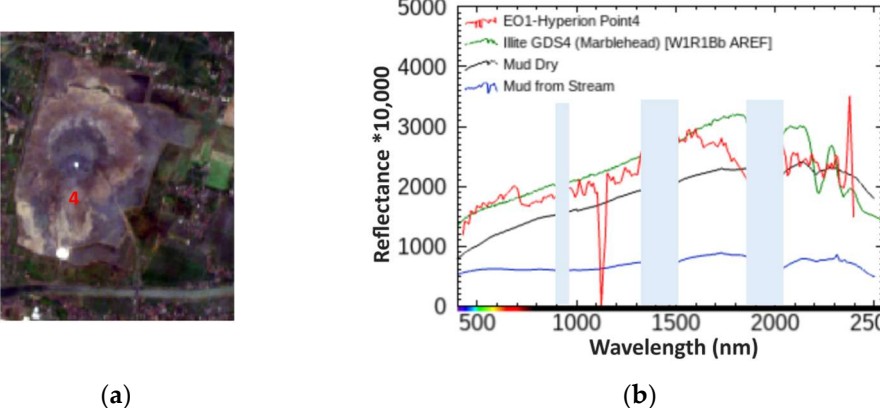

(**a**)          (**b**)

**Figure 11.** (**a**) EO1-Hyperion data natural color composition. (**b**) The EO1-Hyperion spectrum was extracted by the pixel corresponding to point 4 Lat 7°31′46.73″ S, 112°42′51.33″ E and compared vs. the e Illite spectrum and the reflectance of the two samples. Noisy EO1-Hyperion bands due to water vapor effect are masked in light blue.

The PRISMA spectrum extracted in point 1 in the image was compared with Calcite, Chlorite, and Illite and did not show any matching; therefore, these curves have not been plotted in the comparative graph (Figure 12b). Point 1 is located in the north-west of Lusi, and in the image, it appears to correspond to a fluid-rich surface. The spectral library was visually explored, and the Water—Montmorillonite spectrum was selected since it shows similar spectral features. A good spectral agreement between the two spectra was found when overlapped, as shown in Figure 12b.

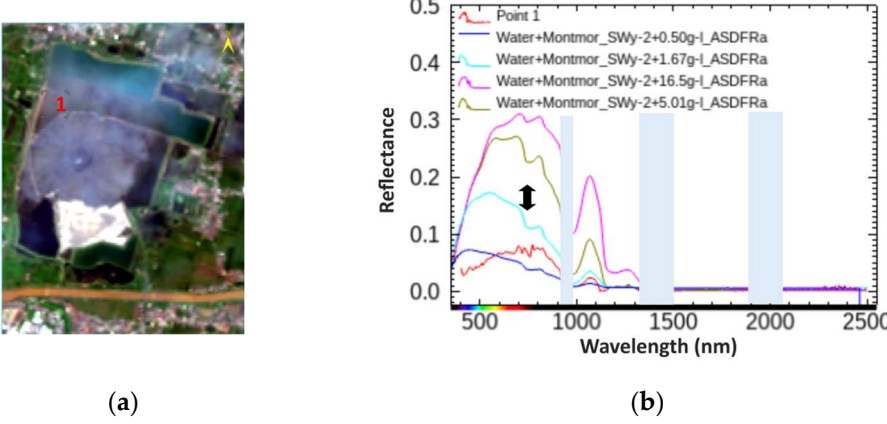

(**a**)          (**b**)

**Figure 12.** (**a**) True color PRISMA data color composition. The PRISMA spectrum (black) was extracted from the pixel corresponding to point 1 and (**b**) compared vs. the spectral library *Water–Mortomolmillite* spectra. Noisy PRISMA bands, due to the water vapor effect, are masked in light blue.

In the USGS spectral library v7, four spectra of Water—Montmorillonite SWy-2 belong to the mineral type Montmorillonite (Smectite). The SWy-2 is a Source Clay Minerals purchased sample from Crook, Wyoming. The Montmorillonite SWy-2, as described by [41], was mixed with distilled water in the proportions of 0.5, 1.67, 5.01, and 16.7 g of montmorillonite per liter of water, and the spectrum measured while the mix was still moving to minimize settling effects. The increase in the Montmorillonite proportion can be seen as an increase in the reflectance values in the 500–1000 nm spectral range. For example, the reflectance maximum value for the SWy-2 0.5 g/L is about 0.07, while it is 0.16 for the Swy-2 1.67 g/L. PRISMA's maximum reflectance value in the same spectral

range (Figure 12b) is about 0.008, suggesting a composition of the Water—Montmorillonite between 0.5 g/L and 1.67 g/L.

The PRISMA spectrum extracted from point 2 was compared with Calcite, Chlorite, and Illite without revealing and matching and, therefore, has not been plotted in the comparative graph (Figure 13b). Point 2 is located in the Southwest part of the Lusi embankment and appears to be water-rich. The spectral library was visually explored, and the seawater spectrum was selected because it showed similar spectral features. A good spectral agreement is shown in Figure 13b. Future work will be carried out to apply techniques that better constrain the composition. In terms of reflectance trend, spectra acquired in point 2 show a good agreement with the seawater spectra. However, at a closer look, the reflectance spectrum shows some spectral features that suggest a mixing composition with a predominant water component.

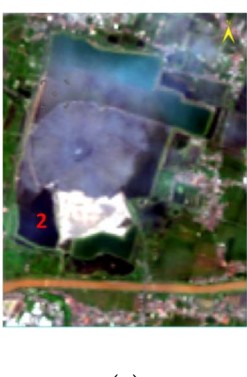

(a)

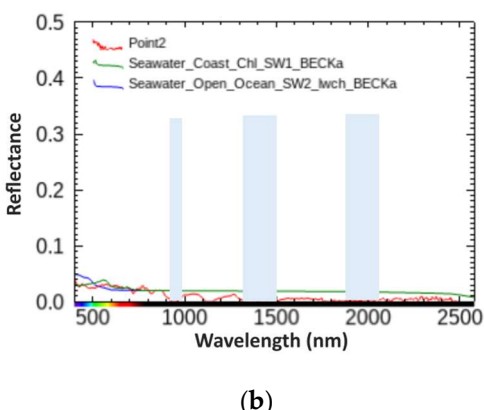

(b)

**Figure 13.** (**a**) True color PRISMA data composition pixel location Lat 7°32′11.18″ S, Lon 112°42′19.54″ E. (**b**) The PRISMA spectrum was extracted by the pixel corresponding to point 2 and compared vs. the spectral library Water–spectra. Noisy PRISMA bands, due to the water vapor effect, are masked in light blue.

Spectra acquired in the very bright white area in the southeast of the Lusi embankment area (point 3) have been compared against the spectra from the spectral library (Figure 14). Calcite appeared to completely mismatch the PRISMA spectrum and was not plotted. The first part of the spectrum acquired in point 3 results in good agreement with Illite Marblehead, while the behavior between 1500 nm and 2500 suggests some mixing effect. Specifically, in the spectral range between 1000 and 1800 nm, the shape of the spectrum is flattened compared with the Illite spectrum in the same spectral range, which reminds the Ammonium–Smectite shape in Figure 3a. The reflectance between 700 and 800 nm appears to be noisy, suggesting that atmospheric effects are still present in the spectrum.

The spectra acquired in point 4 (Figure 15b) show a flattening effect between 1500–1800 nm and 2000–2300 nm. Absorption features of OH, $H_2O$, and AL-OH can be distinguished in the PRISMA spectrum. However, the reflectance between 700 and 800 nm appears to be noisy, suggesting that atmospheric effects are still present in the spectrum.

The PRISMA reflectance (in red) acquired close to the main crater (point 5) has been compared versus the spectral library (Figure 16b). Note that the reflectance values are lower than the previously acquired spectra, and the overall shape suggests the presence of fluids that are in agreement with the visual inspection of the true color. We can notice that the PRISMA reflectance between 700 and 800 nm appears to be noisy, suggesting that atmospheric effects are still present in the spectrum.

Figures 17a and 18a show EO1-Hyperion and PRISMA after applying the EFFORT tool and the continuum removal. The spectra are compared to the continuum-removed reflectance spectra of the samples. Figures 17b and 18b show the spectra in the SWIR spectral range.

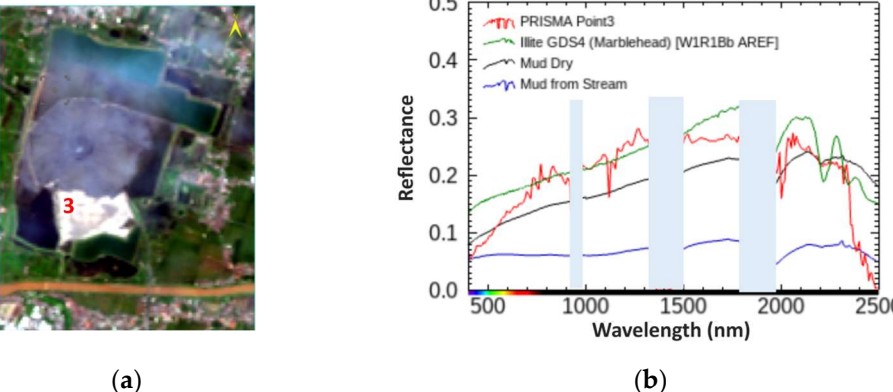

(**a**)                                   (**b**)

**Figure 14.** (**a**) True color PRISMA data color composition., pixel location Lat 7°32′20.5″ S, Lon 112°42′34.67″ E (**b**) The PRISMA spectrum was extracted by the pixel corresponding to point 3 and compared vs. the two samples and the Illite spectrum. Noisy PRISMA bands, due to the water vapor effect, are masked in light blue.

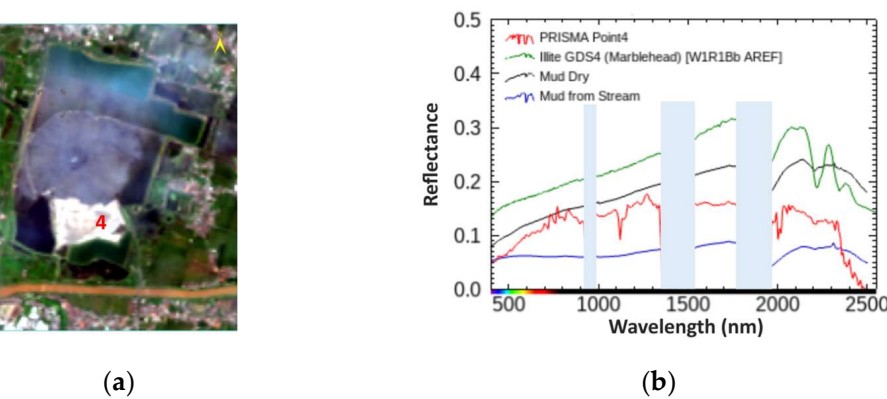

(**a**)                                   (**b**)

**Figure 15.** (**a**) True color PRISMA data color composition. (**b**) The PRISMA single-pixel spectrum was extracted by the pixel corresponding to point 4 and compared vs. the Illite from the spectral library and the two samples. Noisy PRISMA bands, due to the water vapor effect, are masked in light blue.

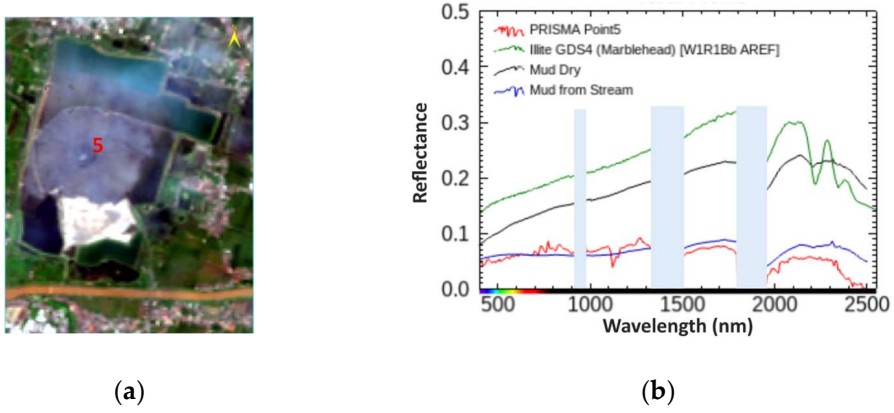

(**a**)                                   (**b**)

**Figure 16.** (**a**) True color PRISMA data color composition. The PRISMA single-pixel spectrum was extracted by the pixel corresponding to point 5 and (**b**) compared vs. the Illite from the spectral library and the two samples. Noisy PRISMA bands, due to the water vapor effect, are masked in light blue.

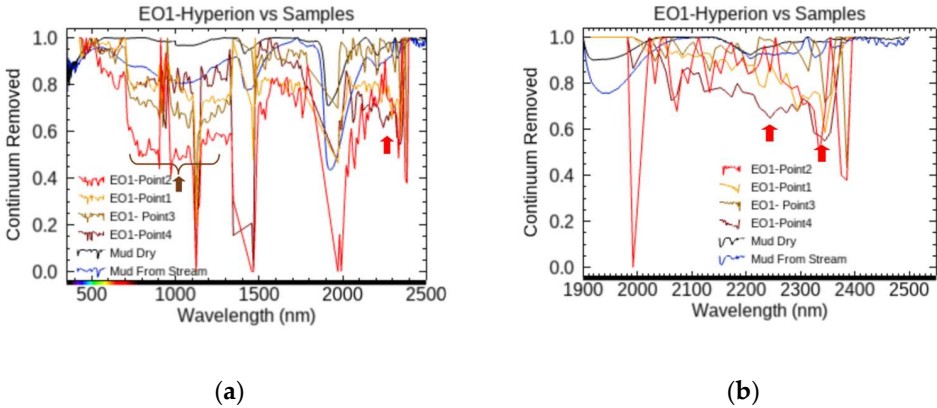

(**a**)  (**b**)

**Figure 17.** (**a**) Continuum removed spectra of EO1 Hyperion vs. *Mud Dry* (black) and *Mud from the Stream* (blue). The brown arrow shows a broad Fe$^+$ absorption band, while the red arrow points at the 2350 absorption Illite band. (**b**) Zoomed continuum removed spectra between 1900 and 2500 nm. The red arrows point to 2200 nm and 2350 nm Illite absorption bands.

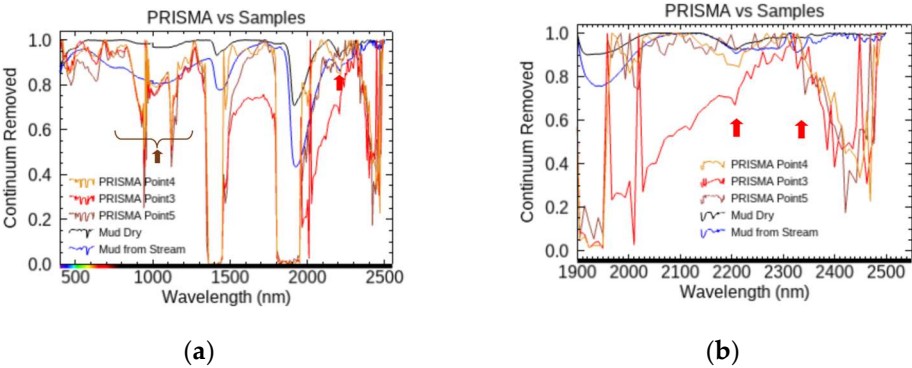

(**a**)  (**b**)

**Figure 18.** (**a**) Continuum removed spectra of PRISMA vs. *mud dry* (black) and *mud from the stream* (blue). The brown arrow shows the broad Fe$^+$ absorption band, while the red arrows point at 2200 nm and 2350 nm Illite absorption bands. (**b**) Zoomed continuum removed spectra between 1900 nm and 2500 nm. The red arrows point at Illite bands.

### *3.3. Comparison EO1-Hyperion-PRISMA*

A walkable area present in both EO1-Hyperion and PRISMA has been identified (ROI) and the mean spectrum has been compared with the samples (Figures 19 and 20).

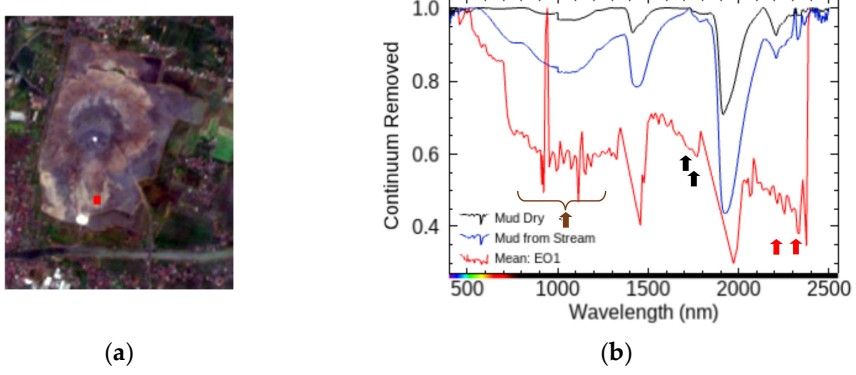

(**a**)  (**b**)

**Figure 19.** (**a**) True color EO1_Hyperion data color composition. The Region Of Interest (ROI) (in red) is located in the South Lusi walkable area. (**b**) Fe$^+$ broad features are indicated by a brown arrow; the two black arrows indicate absorption bands at 1719 nm and 1779 nm, respectively, very close to the hydrocarbon doublet; the red arrows show Illite (2200 nm and 2350 nm) absorptions.

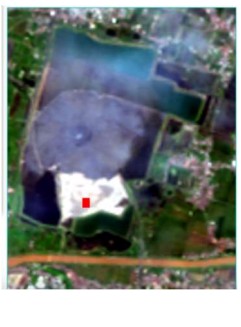
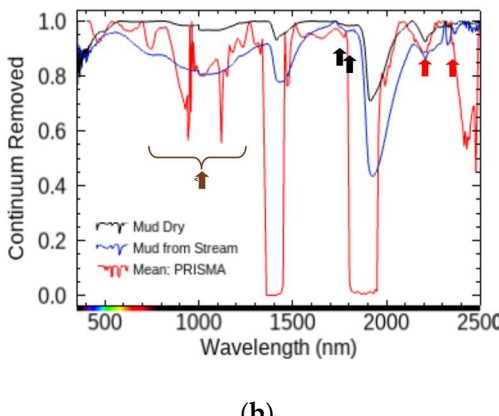

(**a**)                                                    (**b**)

**Figure 20.** (**a**) True color PRISMA data color composition. The Region Of Interest (in red) is located in the South Lusi walkable area. (**b**) Continuum removal spectra of PRISMA and *mud dry* (black) and *mud from stream* (blue). The brown arrow shows the $Fe^+$ broad absorption feature; the two black arrows indicate absorption bands at 1719 nm and 1779 nm, respectively, very close to the hydrocarbon doublet; the red arrows show Illite (2200 nm and 2350 nm) absorptions.

Figure 21 shows the result for the Spectral Angle Mapper classification applied at EO1-Hyperion and PRISMA by using the selected spectra and adding the vegetation and urban area reflectance spectra.

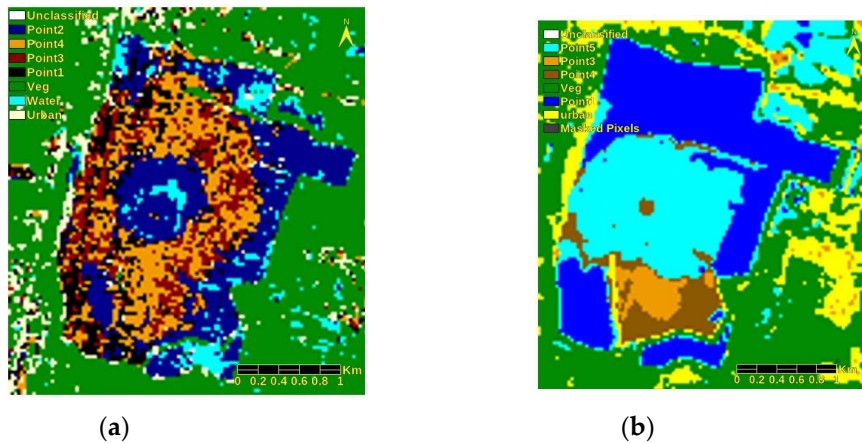

(**a**)                                                    (**b**)

**Figure 21.** (**a**) EO1-Hyperion classification map shows the distribution of different classes. Illite features were found in the classes Point 1 and Point 4. (**b**) PRISMA classification map: the classes point 1 and point 2, mainly water, were merged. Illite and possible Chlorite features are associated with the classes point 3 and point 4.

## 4. Discussion

This study explores the surface properties of reflectance data from samples collected at the Lusi eruption site and compares them with satellite images acquired from the same locality. The Lusi surface has been explored at both laboratory and space scales. The spectra from satellite EO1-Hyperion and PRISMA have been analyzed in order to identify minerals from space. The comparison between the spectral library reflectance and the reflectance of the two Lusi mud samples resulted in the match of the *mud dry* spectra with the Illite GDS4 Marblehead and the spectra of the *mud from the stream* consistent with Illite IMt-1.a W1R1Bb AREF. By comparing the composition of the samples with the spectral library, we found them to be very similar, confirming the ability to derive material composition by using reflectance. The absorption features related to Al-OH, $H_2O$, and OH were distinctive in both samples. The analyzed clayey fraction of the mud samples

is dominated by $SiO_2$ and $Al_2O_3$ and has a high content of other major oxides, such as $Fe_2O_3$, $Na_2O$, and $K_2O$, indicating an Illite/Smectite-rich clay. This is also confirmed by the XRD analysis conducted by the USGS [48], which shows that the clayey fraction of the Lusi mud is dominated by Smectite, Illite, and mixed Smectite and Illite over Kaolinite and Chlorite minerals [13,49]. Sampled mud could come from the volcano-clastic layer or mixed with material from the Smectite-rich stratum between 1341 and 1432 m during the uprising towards the surface. Although the mud analyzed during the initial phases of the eruption revealed strong components originating from the illitized Bluish Grey clay from the Upper Kalibeng Formation [13], more recent analyses also revealed the input of the ~4 km deep shales from the Ngimgbang Formation that are strongly illitized [21,46,50].

EO1-Hyperion satellite versus spectral library shows the ability to pick up overall shapes of the spectrum, as shown in Figures 8 and 10, allowing the identification of the mineral and water (Figure 8). The overall spectra were found to be similar to Illite GDS Marblehead. However, low reflectance surfaces and low SNR can challenge the ability to derive the composition without some additional processing. An improvement was achieved by applying the noise-removing EFFORT technique. The continuum removal (Figures 17 and 19) allowed a better characterization of the $Fe^+$ absorption broadband and the doublet associated with Hydrocarbon. The Illite bands appear weak and in a noisy part of the spectrum.

The better SNR of PRISMA can be seen in a reflectance spectrum that appears to have limited noise. The overall spectral shape of the PRISMA reflectance allows it to identify main components and some distinctive absorption bands. However, systematic higher reflectance values in the spectral range (700–800 nm) have been noticed in our L2D data, suggesting that they are linked to atmospheric correction. The overall spectral shape of the PRISMA reflectance allows us to identify its distinctive absorption band at 2200 n). The continuum removal allowed a clear identification of the $Fe^+$ broadband components and to better distinguish the bands in the SWIR range (Figures 18 and 20). A good agreement was found between PRISMA reflectance and spectral features in correspondence of the walkable area (Figure 20b) vs. dry mud sample. The absorption band at 2350 nm was not evident in both spectra.

A preliminary classification using Spectral Angle Mapper (SAM) was realized using the selected spectra with the addition of the class vegetation and urban. Figure 21a,b show the distribution of the different classes. Although the PRISMA data were acquired in the "dry season" (March–September), walkable areas are limited in the PRISMA image. The spectrum acquired by both EO1-Hyperion and PRISMA satellites has a spatial resolution of 30 m (i.e., covers an area of 900 $m^2$). If the surface is not homogenous, the measured spectrum is a mix of different materials, which can influence the composition retrieval. In those cases, some alternative techniques (i.e., unmixing) would be considered. This study demonstrates the potential of the PRISMA sensor to identify spectral features and mineral components of clastic eruptive features such as Lusi. Furthermore, the use of hyperspectral sensor at improved SNR, SWATH dimension, and GSD improvement using pan sharpening techniques like PRISMA has the potential to help provide useful information to determine the mineral composition and may provide insights regarding the system evolution and changes in mineralogical composition through time. Additional processing of the PRISMA data, unmixing techniques, and machine learning will be used to evaluate the potential of composition mapping in such a challenging environment. New hyperspectral space sensors, such as PRISMA and the recently launched EnMap and EMIT, offer the opportunity to investigate and monitor surface composition in remote areas at increased repetition time, allowing the evolution of the observed area to be mapped in time. Furthermore, the data acquired by EO1-Hyperion and PRISMA can provide a synoptic view augmented by their unique spectral ability, which allows for a map of areas of Lusi that are difficult to reach during fieldwork, promising to be of great support and complementarity. Finally, the spectral characterization from satellite systems relies on the sensor's good performance (i.e., SNR, spectral resolution, and calibration) and reliable atmospheric correction. Although the

applied EFFORT technique reduced the noise, the L2 PRISMA product still shows some systematic noise in the VNIR (750–850 nm) where some mineral diagnostic features occur, potentially leading to misinterpretation. For future mineral mapping applications, we suggest retrieving the reflectance starting from the PRISMA L1 radiance and applying a more specific atmospheric correction. Although running systematic spectroscopic field campaigns would be ideal for validating the satellite data, laboratory measurements on samples acquired near the satellite passage are a viable compromise to consider due to the high cost and difficulties of carrying out such fieldwork.

## 5. Conclusions

This study describes a multidisciplinary approach to investigate the reflectance spectra of mud erupted at the Lusi vent (east Java, Indonesia) and compare them with data acquired from satellites EO1-Hyperion and PRISMA imaging the same site and ultimately with those available from the spectral library databases (USGS spectral library version 7).

The comparison between the spectral library and the Lusi samples shows a good agreement with the oxides represented in high percentages (i.e., $SiO_2$, $Al_2O_3$, and $Fe_2O_3$). Some low percentages of oxides (i.e., $MnO$ and $K_2O$) show differing amounts when comparing the Lusi *mud collected from the stream* and the Illite IMt-1.a from the spectral library sample. Similar behavior is detected in the *dry mud* sample (i.e., $SiO_2$, $Al_2O_3$, $MnO$, and $K_2O$) compared to the Illite GDS4 Marblehead, except for $Fe_2O_3$ present in a higher percentage.

PRISMA spectra acquired 7 years later; the samples show features typical of Illite VNIR Fe-electronic feature and absorption in the SWIR (2200 nm) comparable with the dry mud reflectance of the sample measured in the laboratory.

The measurements completed by Hyperion and PRISMA L2D show the ability of the sensors to characterize spectra associated with specific minerals that can be related to the Lusi mud composition. Overall, better atmospheric correction is needed to remove artificial features that could negatively influence the identification of spectra. It also allows for reliable retrospective analysis and the derived material compositions.

The main Illite absorption bands were identified by using PRISMA reflectance. The continuum removal showed the Fe+ broad absorption and AL-OH at 2200 nm.

Overall, this study reveals the potential of hyperspectral technology from space to identify materials in challenging environments that are also influenced by large variability in terms of compositions, water content, and gas emissions. Identifying weak absorption features is possibly challenged by the Atmospheric correction implemented in the PRISMA L2 product. The increasing availability of hyperspectral image archives and time series in synergy with airborne acquired data will improve (by using deep learning will greatly improve the analysis of surface mineral compositions from space and their evolution in the presence of geophysical phenomena and manmade activities.

**Author Contributions:** Conceptualization, S.A.; methodology, S.A.; formal analysis, S.A. and A.S.; writing—original draft preparation, S.A., M.F.B. and A.S.; writing—review and editing, S.A., A.S. and A.M. All authors have read and agreed to the published version of the manuscript.

**Funding:** PRISMA Products © of the Italian Space Agency (ASI), delivered under an ASI License to use. The Italian Space Agency is thanked. USGS spectral library and EO1-Hyperion are thanked for providing the data at a free-of-cost policy. The work was supported by the European Research Council under the European Union's Seventh Framework Programme Grant agreement no. 308126 (LUSI LAB project, PI A. Mazzini). We acknowledge the support from the Research Council of Norway through its Centres of Excellence funding scheme, Project Number 223272 (CEED) and the HOTMUD project (288299).

**Data Availability Statement:** The original contributions presented in the study are included in the article, further inquiries can be directed to the corresponding author.

**Conflicts of Interest:** The authors declare no conflicts of interest.

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
