# Peer review of "Mud Spectral Characteristics from the Lusi Eruption, East Java, Indonesia Using Satellite Hyperspectral Data"

_geosciences, doi:10.3390/geosciences14050124_

Round 1
Reviewer 1 Report
Comments and Suggestions for Authors
Dear authors I find this Technical note quite interesting for readers both for the geological setting (the mud volcano complex of Lusi) as well as on the use of heterogeneous hyperspectral data.
As you rightly point out, the application of hyperspectral data from satellite platforms to challenging and time-varying environments can provide particularly effective answers, especially for time-series potential.
I only have a few notes to make with respect to the way you have set up the article:
-- first of all, the quality of some of the figures (especially 2a and 2b, 5b, 6b, 13b, 14b, 15b, 16b, and 17b) for the spectral part is not optimal. I recognize that it is never easy to represent this data (the spectral ones), but you should in my view be able to enlarge the spectral boxes to make them more readable and so useful.
-- secondly, I wonder about the real usefulness of chap 2.4 "Examination of hyperspectral sensors reflectance spectra". In this part, the authors show several spectral behaviors, both for the EO1-Hyperion and PRISMA, but I can't understand how it relates to the other parts of the article. Moreover, even in its development, the intent is unclear. Would you like to demonstrate the increase in information retrievable from PRISMA compared to EO1? And why are the samples used for the two sensors not the same? Of course, I understand that six years in such a dynamic environment can totally change the landscape and materials. Anyway, I suggest removing it or explaining better the flowchart (right?) of work. Furthermore, probably the 2.4 chap it's better after the description of Sensor (chap 2.2.1 & 2.2.2).
-- thirdly, I would add some information on how you matched spectral libraries and signatures collected from hyperspectral measurements in the laboratory and from satellite.
To follow some minor annotations on the text:
rows 52-53 "..the main sources of the Lusi water seawater,..." unclear
row 110 "..ENVI software which allows to remove the overlapping wavelength as well." what's means overalapping wavelength?
row 134 remove "are called spectral libraries"
row 175 "..by some alterations." Please add some info about these alterations.
row 182 "..(about 5cm diameter each)..". This info, so specific, seems unuseful to me at least in this context, don't you?
rows 187-189 "The measurements were repeated five times in different points of the sample. Reflectance was measured in three different points in the dry mud sample (Figure 3) due to the smaller size of the sample." --> it's unclear how this is rappresented in Fig.4. Seems to me, from the pictures that MUDdry (the dried one?) was sampled 5 times and not 3, and the MUDfromstream 3 times. Please clarify.
row 332 "..Clay Minerals purchased sample from..", what's mean purchased in this phrase?
row 448 remove open parenthesis from "..(EO1-Hyperion and PRISMA..".
Author Response
Dear anonymous reviewer,
Thank you for your suggestions, comments and time in reading and reviewing our technical note note Please find attached the answers.
Best wishes
Stefania

Reviewer 2 Report
Comments and Suggestions for Authors
This paper seems lost in what it is trying to achieve. It could be about how 20 years of technological development has advanced the ability of satellite-based hyperspectral (VNIR-)SWIR sensors (i.e., Hyperion launched in 2000 versus PRIMSA launched in 2019) to better measure the often small but diagnostic absorption features of minerals like illite (~2200, ~2350 nm features) and chlorite (2000, ~2250 and ~2330 nm features), even in a challenging “wet” environment like the Indonesian mud volcanoes. But the paper fails to show a single satellite pixel spectrum (let alone a mineral map givenb PRISAM is an imaging sensor) that shows a clear mineral SWIR absorption.
Suggest the authors take a look at the first papers published on Hyperion pixel mineral spectra, especially those from the test sites at Mt Fitton and Panorama in Australia (IGARSS 2001 and 2002). These studies show that even the low SNR Hyperion sensor can measure and map small/subtle SWIR absorptions (across the entire 2150 to 2350 nm region) caused by minerals like muscovite (with different wavelengths driven by levels of Tschermak substitution), pyrophyllite, kaolinite, chlorite, talc, tremolite and dolomite. One of the papers from Mt Fitton also shows that the same pixel spectra are recorded by Hyperion from the same ROI over nine separate satellite overpasses, showing both clear mineral absorption measurement integrity and that much of the pixel-spectrum noise trhat plagues the Hyperion and PRIMSA presented by the authors is systematic and hence removable, which is also presented shown in related papers.
A few other points:
(i) Associated field validation show illite absorption at 2200 nm (Figure 8a and 8b). Cannot see any field validation spectra showing chlorite SWIR (not broad Fe-related NIR absorptions) absorptions. Is this the case? Not clear from the too small/congested figures.
(ii) All Figures presenting the satellite pixel spectra are overshadowed by too many and unnecessary USGS library spectra. Your field validation spectra are best as these are closest to the “truth” that could be measured using Hyperion/PRISMA.
(iii) The satellite pixel spectra are riddled with noise. Try EFFORT suppression anmd maybe look at removal of systematic detector noise in Mt Fitton Hyperion data (IGARSS 2002). Also, Figures are not optimised in the y-axis for the target satellite pixel spectra (hull quotients (continuum-removal) would be better for all data types) .
(iv) Airborne hyperspectral VNIR-SWIR is not small-scale and is not just AVIRIS. The HyMap sensor has conducted many public (up to 250,000 km2 each for most of the geological surveys of Australia as well as the country-wide survey of Afghanistan by the USGS) and private (e.g. DeBeers surveys across much of Africa, India and Australia) regional surveys that more than compare in size to what has been collected to date by the new suite of satellite hyperspectral VNIR-SWIR sensors.
(v) How can you say that PRISMA is opening up “new opportunities….to characterize minerals in semi-arid environments”. Please delve deeper into the literature regards hyperspectral mineral mapping published over the last 40 years.
(vi) Illite can take water molecules into its interlayers and in the process also swell, although nowhere near as much as form example montmorillonite, but indeed much more so than kaolinite.
(vii) Illite does not transform from/to smectite. Illite formation requires a source of K.
(viii) What is “poorly-hydrated potassium ions?
(ix) 1400 nm feature is also caused by water.
(x) Smectites have either a dioctahedral or trioctahedral structure. Trivalent cations are typical of the former whereas divalent cations are typical of the later. Saying that smectites can have a significant amount of Mg or Fe is too simplistic to be meaningful.
(xi) Not sure what value the oxide geochemistry provides.
(xii) Use wavelength zoom to show target mineral absorptions.
(xiii) Cull the most aberrant noise in the spectra (especially around the edges of the water vapour bands at 860, 1140, 1440 and 1940 nm) or maybe do not use tie-lines between Hyperion/PRIMSA data points as all you end of seeing is the noise.
(xiv) Are you sure the Hyperion and PRISMA scene acquisitions were from the driest times possible? When did it rain last for each date of collect?
(xv) Not sure of the value of wet montmorillonite and sea water spectra, which only show low albedo and water features in the VNIR (Figure 19).
(xvi) There is no reminding of NH4 features in Figure 15. The broad absorption between 1000-1800 nm for the USGS library spectra (in Figure 2a not Figure 3a) is related to Fe2+ (library mineral spectra are not always pure). The NH4 in illite produces a narrow feature in the 2020-2115 nm region which is not apparent in your satellite pixel spectra.
(xvii) Japan’s HISUI not mentioned.
Finally, focus your effort on what you are trying to achieve. Suggest it is the ability of hyperspectral satellites to reliably measure SWIR mineral absorption features. It is paramount that your figure/s clearly show that at least PRISMA can measure the diagnostic absorptions of illite [2200 nm (major) and 2350 nm (minor)], which is expected given the field validation spectra. Chlorite (2000, 2250 and 2330 nm) would also be good but not clear if is present given the available field validation spectra. If illite measurement (at east) is achievable then please also generate a related mineral map to show the spatial distribution around the mud volcano.
In the end, I suspect your selection of a “wet” site was too challenging given “wetness” greatly suppresses the SWIR-2 (your Figures 13 and 14).
Comments on the Quality of English LanguageEnglish expression is good.
Author Response
Dear reviewer, thank you for your time in reading and reviewing our technical note. We appreciate your precious comments. Please find attached the answers to the comments.
Best wishes
Stefania

Round 2
Reviewer 1 Report
Comments and Suggestions for Authors
Dear authors,
the text is much improved, but there are still problems with the graphics, and I have a doubt that it is related to the editing-formatting stage.
Overall and I repeat, only for the figures, I find in the pdf, some problems with scale and consequent readability.
I list the main problems I am experiencing reading the paper-draft:
fig.1 the complex it's not center on the page
fig.2 the inset A has a different size from the B. Scale to B?
fig.7 the inset B is partially clipped by the page margin
fig.16 like fig1
fig 17 and 18 reading these important figures is difficult at this scale.
fig. 19 and 20 enlarge the two insets.
Best
Author Response
Dear reviewer.
Thank you for your second review of our article – we are glad for you noticing this, our efforts to improve and also improve the quality of the article and we are happy to hear that the changes are visible. Here attached we are answering your last comments to our article and hopefully it will meet your expectations.
Thank you again for every valuable comment that you provided us for the last two rounds of reviews.
We wish you a nice rest of the day.
Best wishes
Stefania and the co-authors

Reviewer 2 Report
Comments and Suggestions for Authors
The figures have been improved to some degree though why the y-axis continues to be scaled 0-1 when the dynamic range of the data is typically <30% is puzzling.
The Intro is better but the airborne context remains mis-represented. Again, it is important to make clear that: AVIRIS is a NASA-research-tool only with severe restrictions for its use outside of the USA, hence the “small area” acquisitions. Because of this restriction, the global scientific and commercial communities have been using (since 1995) other airborne sensors from HyVista, ITRES and SpecTIR (+others) to routinely collect hyperspectral VNIR-SWIR data for “mineral and geological characterization, especially for “large areas” surveys (including at the scale of countries and continents).
You have a noise problem. Again, a simple EFFORT correction will remove most of the systematic stuff, including the CO2 features around 2000 nm and the spikes and drop-down >2350 nm plus the (Figures 18 and 19).
Figures 18 and 19 are very useful (well done) but please compare the satellite spectra with your field spectra but not the USGS library spectra.
The sharp 3-band feature at ~2000 nm is not NH-4 in illite. If it was, then: (i) did you measure it in the field samples but failed to include it in the Figure 4; (ii) why is the shape so sharp in Figure 19b (3-bands only) and not like the V-shape shown by the USGS spectra in Figure 2; (iii) why are not the more pronounced 2100 NH4 and the dominant illite 2200 nm absorption also present and more much deeper than this secondary-NH4 2000 nm feature that overlaps with and looks like CO2 absorption.
Readers will not be impressed by the only the poor expression of a 2200 nm AlOH feature (2350 nm is not evident even in your field spectra, which is not a surprise as we have found the same for low-temp illites), which could instead be caused by any AlOH-bearing mineral. No enhanced mineral mapping demonstrated here above that achievable using multispectral ASTER.
In the end, this is not the standard of mineralogical detail we (the community) have been achieving routinely around the world using operational airborne VNIR-SWIR-TIR hyperpsectral systems (and not AVIRIS).
Finally, DESIS is hyperspectral VNIR sensor - i.e., does not have SWIR bands and so cannot do AVIRIS-type mineral mapping. If you include DESIS then there are other many hyperspectral VNIR airborne and satellite sensors to include but this also then changes your focus away from the SWIR hydroxyl vibrations which may be a good thing at least for your wet mud test site. That is, change your focus for this wet mud test site from "narrow SWIR cation-hydroxyl-vibration absorptions" to the more achievable/convincing broad VNIR Fe-electronic features.
Author Response
Dear Reviewer
Thank you very much for your feedback.It is very appreciated as it helping to improve the standard of the manuscript.
Here below the answers to the new comments we have received.
Q .The figures have been improved to some degree though why the y-axis continues to be scaled 0-1 when the dynamic range of the data is typically <30% is puzzling.
A. We choose the range as typically the reflectance are plotted between 0-1.The suggestions from the reviewer have been implemented and all the reflectances graphs have been plotted at 50% reflectance. This allows to appreciate both absorption features and overall shape of the spectra.
Q. The Intro is better but the airborne context remains mis-represented. Again, it is important to make clear that: AVIRIS is a NASA-research-tool only with severe restrictions for its use outside of the USA, hence the “small area” acquisitions. Because of this restriction, the global scientific and commercial communities have been using (since 1995) other airborne sensors from HyVista, ITRES and SpecTIR (+others) to routinely collect hyperspectral VNIR-SWIR data for “mineral and geological characterization, especially for “large areas” surveys (including at the scale of countries and continents).
A. We have expanded this part naming the airborne hyper spectral systems used for mineralogy mapping. A reference was also added.
Q. You have a noise problem. Again, a simple EFFORT correction will remove most of the systematic stuff, including the CO2 features around 2000 nm and the spikes and drop-down >2350 nm plus the (Figures 18 and 19).
- Thank you for the information provided. We have been using the EFFORT tool available under ENVI on both PRISMA and Hyperion and re-done all the graphs and the figures. An improvement was noticed.Thanks for suggesting this polishing methods I will be using it in the future.
Q. Figures 18 and 19 are very useful (well done) but please compare the satellite spectra with your field spectra but not the USGS library spectra. Including Fig 18 and 19 that are now compared vs in situ.
A. The spectra in Figure 18 and 19 have been compared with the in situ samples.
Q The sharp 3-band feature at ~2000 nm is not NH-4 in illite. If it was, then: (i) did you measure it in the field samples but failed to include it in the Figure 4; (ii) why is the shape so sharp in Figure 19b (3-bands only) and not like the V-shape shown by the USGS spectra in Figure 2; (iii) why are not the more pronounced 2100 NH4 and the dominant illite 2200 nm absorption also present and more much deeper than this secondary-NH4 2000 nm feature that overlaps with and looks like CO2 absorption.
Readers will not be impressed by the only the poor expression of a 2200 nm AlOH feature (2350 nm is not evident even in your field spectra, which is not a surprise as we have found the same for low-temp illites), which could instead be caused by any AlOH-bearing mineral. No enhanced mineral mapping demonstrated here above that achievable using multispectral ASTER.
In the end, this is not the standard of mineralogical detail we (the community) have been achieving routinely around the world using operational airborne VNIR-SWIR-TIR hyperpsectral systems (and not AVIRIS).
- The aim of the this technical note is not to impress the reader it aims to evaluate the L2 PRISMA reflectance product, to see if some mineral information can be retrieved when we look at challenging environment (wet- muddy). We don’t expect to achieve the same standard performances that you gain from airborne with better spatial and temporal resolution. However, the availability of several hyperspectral systems from space will open new opportunity. Next research would explore the synergy of the two different scale systems by using training approaches (deep learning). Finally and personally I believe that understanding better how the “ low standard “ of satellite systems are related to the “airborne high standard” reached by the community will allow to improve the understanding of the resources available on other solar system bodies that have been made possible by the hyper spectral sensors on orbiting probes from the last few decades and that are still possible mainly through orbiting systems.
Q.Finally, DESIS is hyperspectral VNIR sensor - i.e., does not have SWIR bands and so cannot do AVIRIS-type mineral mapping. If you include DESIS then there are other many hyperspectral VNIR airborne and satellite sensors to include but this also then changes your focus away from the SWIR hydroxyl vibrations which may be a good thing at least for your wet mud test site. That is, change your focus for this wet mud test site from "narrow SWIR cation-hydroxyl-vibration absorptions" to the more achievable/convincing broad VNIR Fe-electronic feature
A. Fully agree. Thank you for the comment it is very valuable. We took it into account in the conclusions.
Round 3
Reviewer 2 Report
Comments and Suggestions for Authors
The authors have sufficiently improved the paper, especially the figures, introduction and conclusions to warrant publication. However, the paper was always a challenge given the wrong choice of test site (too wet and limited SWIR-active minerals). If they wanted to compare PRISMA with Hyperion then a dry site with a better range of AlOH and MgOH minerals would have been more appropriate.
Author Response
Dear reviewer.
Thank you for your third review of our article – we are glad for you noticing this, our efforts to improve and also improve the quality of the article and we are happy to hear that the changes are visible. Hereby we are answering your last comment to our article and hopefully it will meet your expectations.
Response to Reviewer’s comment:
Comment: The authors have sufficiently improved the paper, especially the figures, introduction and conclusions to warrant publication. However, the paper was always a challenge given the wrong choice of test site (too wet and limited SWIR-active minerals). If they wanted to compare PRISMA with Hyperion then a dry site with a better range of AlOH and MgOH minerals would have been more appropriate.
Responce:
We agree with the reviewer that, for a comparison of the two sensors, a dry site would be more appropriate. We hope that with new sensors and retrospective analysis of Hyperion, the selection of a dry site in multiple images would be while making possible also a monitoring of the Lusi evolution.
Thank you again for every valuable comment that you provided us for the last two rounds of reviews.
We wish you a nice rest of the day.
Best wishes
Stefania and the co-authors